# Piecewise Modeling the Accumulated Daily Growth of COVID-19 Deaths: The Case of the State of São Paulo, Brazil

**DOI:** 10.3390/e23081013

**Published:** 2021-08-04

**Authors:** Erlandson Ferreira Saraiva, Carlos Alberto de Bragança Pereira

**Affiliations:** 1Institute of Matematics, Federal University of Mato Grosso do Sul, Campo Grande 79070-900, Brazil; 2Institute of Matematics and Statistics, University of São Paulo, São Paulo 05508-090, Brazil; cadebp@gmail.com

**Keywords:** COVID-19, growth model, piecewise model, estimation, non-linear minimum square

## Abstract

The pandemic scenery caused by the new coronavirus, called SARS-CoV-2, increased interest in statistical models capable of projecting the evolution of the number of cases (and associated deaths) due to COVID-19 in countries, states and/or cities. This interest is mainly due to the fact that the projections may help the government agencies in making decisions in relation to procedures of prevention of the disease. Since the growth of the number of cases (and deaths) of COVID-19, in general, has presented a heterogeneous evolution over time, it is important that the modeling procedure is capable of identifying periods with different growth rates and proposing an adequate model for each period. Here, we present a modeling procedure based on the fit of a piecewise growth model for the cumulative number of deaths. We opt to focus on the modeling of the cumulative number of deaths because, other than for the number of cases, these values do not depend on the number of diagnostic tests performed. In the proposed approach, the model is updated in the course of the pandemic, and whenever a “new” period of the pandemic is identified, it creates a new sub-dataset composed of the cumulative number of deaths registered from the change point and a new growth model is chosen for that period. Three growth models were fitted for each period: exponential, logistic and Gompertz models. The best model for the cumulative number of deaths recorded is the one with the smallest mean square error and the smallest Akaike information criterion (AIC) and Bayesian information criterion (BIC) values. This approach is illustrated in a case study, in which we model the number of deaths due to COVID-19 recorded in the State of São Paulo, Brazil. The results have shown that the fit of a piecewise model is very effective for explaining the different periods of the pandemic evolution.

## 1. Introduction

Since the discovery of a new coronavirus in the city of Wuhan, China, the virus has spread rapidly worldwide, and created a major global health crisis [1]. In fact, this disease is one of the main health problems in the world and has had an enormous economic and social impact, leading to an increase in poverty and the loss of thousands of lives in many countries. Until 27 April 2021, four million deaths had been registered in the whole world (www.worldometers.info/coronavirus, accessed on 23 July 2021).

This pandemic scenery has increased the interest in statistical models capable of projecting the evolution of the disease in countries, states, or cities. This interest lies mainly in projections that can assist government agencies in making decisions regarding the intensification of social isolation, the acquisition of hospital equipment, an increase in the number of intensive care units in hospitals.

Especially in the year 2020, many articles were published describing modeling procedures for the number of cases and/or deaths due to COVID-19 in many countries. In general, the published works model the accumulated number of cases (or deaths) by using some non-linear growth model. For example, Ref. [2] considered a simple exponential growth model to analyze the initial phase of the epidemic of COVID-19 in Africa, Ref. [3] modeled the data from Philippines and Taiwan using the generalized logistic model, Ref. [4] applied the Richards growth model to the data collected in China, France, Germany, Iran, Italy, South Korea, and Spain, Ref. [5] calibrated the logistic growth model, the generalized logistic growth model, and the generalized Richards model for the number of cases recorded in China, Ref. [6] considered the Gompertz model to predict the number of deaths in Brazil, Ref. [7] estimated the number of total COVID-19 cases and deaths in the UK, Russia, and Turkey using the Gompertz model, among others.

As discussed by the authors cited above, the growth models present satisfactory performance for modeling the accumulated number of cases (or deaths) of COVID-19. However, these models are generally adequate to model data from a specific period of the pandemic. Since the growth in the cumulative number of cases and deaths by COVID-19 has, in general, presented a heterogeneous evolution over time, this implies that the adjustment of only one of these growth models may not be adequate to explain the entire study period.

In order to overcome this issue, this article introduces a piecewise model in which different growth functions are adjusted for different pandemic periods of time. Our proposed modeling is based on the fit of three growth models, exponential, logistic and Gompertz, for each period of the pandemic. Our preference for these three models is based on studies described in the literature that indicate they are excellent for use in longitudinal data, see for example [8,9] and the references therein. However, the proposed modeling is not restricted to these three growth models and can be easily adapted for other growth models.

To identify the different periods of the pandemic in the course of time, we consider a grid of values with increments of size 1, in which, each point of the grid represents a possible change point of the pandemic. The increment of one unit represents the increase of one day in the pandemic. Then, we fit the three growth models for two sub-datasets obtained according to each possible change point. To choose the best model for each sub-dataset, we consider as a criterion the mean square error (MSE), the Akaike Information criterion [10,11], denoted by AIC, and the Bayesian Information criterion [12], denoted by BIC. The best change point *d* is that associated with the model with the lowest mean square error.

The fitting procedure of the three models is based on obtaining the point estimates for the parameters of the growth models by using the non-linear least square method, as described by [13,14]. We have used the software R [15] and the command *nls* of the package *nlstools* [16,17]. Based on the fitted model, we find the coordinates of the inflection point and consequently the estimated date for the peak of the pandemic in the current period.

The proposed modeling is illustrated in a case study, in which we model the cumulative number of deaths recorded in the state of São Paulo, Brazil. However, the modeling may be easily adapted for datasets from other countries, states, or cities. We opt for modeling the cumulative number of deaths instead of the cumulative number of recorded cases because these values do not depend on the number of diagnostic tests performed. This allows us to identify the changes in the pandemic’s evolution without having to adjust for changes in the number of diagnostic tests. On the other hand, one can argue that the modeling procedure now relies on accurate reporting of the mortality statistics due to COVID-19. Since the health secretary of São Paulo state publishes the mortality statistics daily and the information is checked by a group of news-press companies, we consider that these published data are adequate for our modeling procedure. As an illustration of the versatility of our proposal, we also present the model for the cumulative number of recorded cases in the State of São Paulo, Brazil.

The two main advantages of the proposed modeling approach are: 1. its effectiveness in describing the different periods of the pandemic; and 2. its capacity to explain each period of the pandemic through the estimates for the epidemiological parameters of interest, such as the growth rate for the number of daily deaths and the date for the occurrence of the peak.

The remainder of the paper is organized as follows: Section 2 presents the statistical model. Section 3 describes the adopted modeling procedure to obtain a piecewise growth model for the cumulative number of recorded deaths in the state of São Paulo, Brazil. In this section, we also present the piecewise model fitted for the cumulative number of cases recorded in the State. Section 4 concludes the paper with the final remarks. Additional details are provided in Appendix A.

## 2. Statistical Model

Consider $D_{t}$ to be the number of deaths due to COVID-19 recorded in the state of São Paulo, Brazil, on the *t*-th day, for t=0,…,n, where t=0 represents the day that the first death was recorded (17 March 2020) and n=410 is the last day considered in the study (27 April 2021). In this period, 96,191 deaths were recorded. This dataset is publicly available on the website www.seade.gov.br/coronavirus/.

Figure 1 shows the recorded $D_{t}$ values, for t=1,…,n. As one can note, the recorded $D_{t}$ values have a high variability which makes the modeling procedure somewhat complicated. The sample standard deviation of the $D_{t}$ values is 237.8288. Due to this, we opt to develop the modeling procedure by considering the cumulative values because these values present a more stable behavior, as can be viewed in Figure 2a. Figure 2b shows the values from Figure 2a in the log-scale.

Thus, let $X_{t}$ be the number of deaths until the *t*-th day, in an accumulated way, i.e.,
$$X_{0}=D_{0}\;\;\;\;\mathrm{and}\;\;\;\;X_{t}=\sum_{i=0}^{t}D_{i},$$
for t=1,…,n.

Then, we assume the following multiplicative growth model:(1)$$X_{t}=h\left(t|\theta \right)\cdot \nu_{t},$$
where h(t|θ) is a nonlinear growth model indexed by the parameter θ (scalar or vector), $\nu_{t}$ is a random error assumed as being generated from a log-normal distribution with mean μ and variance $\sigma^{2}$, for t=1,…,n.

To complete the model specification, we need now to set up the nonlinear growth model h(t|θ) in Equation (Equation 1), for t=1,…,n. Hereafter, we consider the following three nonlinear growth models: exponential, logistic and Gompertz. Details on these three growth models are presented in Appendix A.

In addition, in order to avoid working with values on the scale of thousands (see Figure 2a), we opt to consider the model Equation (Equation 1) on the logarithmic scale. Thus, by taking the logarithm transformation on both sides of Equation (Equation 1), we have that:(2)$$Y_{t}=log\left(X_{t}\right)=f\left(t|\theta \right)+\varepsilon_{t}$$
where f(t|θ)=log(h(t|θ)) is the logarithmic transformation of a nonlinear growth model h(t|θ) indexed by the parameter θ and $\varepsilon_{t}=log\left(\nu_{t}\right)$ for $t=1,\ldots ,n_{d}$. Thus, from log-normal properties we have that $\varepsilon_{t}$ is a random error assumed as being generated from a normal distribution with mean μ and variance $\sigma^{2}$, $\varepsilon_{t}∼N\left(\mu ,\sigma^{2}\right)$. In addition, we assume that μ=0 and $cov(\varepsilon_{t},\varepsilon_{t^{\prime }})=0$, for $t,t^{\prime }=1,\ldots ,n$ and $t\neq t^{\prime }$.

However, since the cumulative number of deaths by COVID-19 has, in general, presented a heterogeneous evolution over time, the adoption of a single function f(t|θ) to explain the entire considered period may not be adequate. Thus, in order to overcome this issue, we consider that the function f(t|θ) is a piecewise function, given by
f(t|θ)=f1(t|θ),for 0<t≤d1;f2(t|θ),for d1<t≤d2;⋮,⋮⋮;fk(t|θ),for t>dk..
That is, we assume that the entire period of the pandemic is divided into *k* different periods, being $f_{j}\left(t|\theta \right)$ a function considered for recorded data in the period $\left(d_{j-1},d_{j}\right]$, for j=1,…,k.

Table 1 shows the mathematical functions considered for $f_{j}\left(t|\theta \right)$, for j=1,…,k. Other than for the log-exponential model that presents an infinite growth, the log-logistic and log-Gompertz have growth limits. The value $\alpha_{1}=log\left(\alpha_{1}^{*}\right)$ is the upper asymptote for both log-models, where $\alpha_{1}^{*}$ is the upper asymptote of the models in the original scale (see Appendixes Equation 8 and Equation 9). In the context of COVID-19, the value of $\alpha_{1}^{*}=\exp\left\{\alpha_{1}\right\}$ is an estimate for the maximum number of deaths that will occur. The parameters $\left(\alpha_{2},\alpha_{3}\right)$ are related to the coordinates I=(T,f(T|θ)) of the inflection point. From a practical viewpoint, the point *I* is the peak of the pandemic, i.e., the point in which the curve of the number of deaths (log-transformed) changes its slope (positive to negative), meaning that before the peak the number of deaths grows at an increasing rate (positive slope) and after the peak, the growth is characterized by a decreasing rate (negative slope). That is, after the peak it is expected that the number of deaths is smaller each day. In addition, we have that *T* is the day on which the peak will occur and f(T|θ) is the logarithm of the cumulative number of deaths that will be recorded until the day *T*.

In order to get the parameter estimates of the three log-growth models, we adopt the nonlinear least square method. For this, we use the software R [15] and the command *nls* of the package *nlstools* [16,17]. Denote the estimates for the parameters of interest θ by $\hat{\theta }$, where $\hat{\theta }=\left({\hat{\alpha }}_{1},{\hat{\alpha }}_{2}\right)$ for the log-exponential model and $\hat{\theta }=\left({\hat{\alpha }}_{1},{\hat{\alpha }}_{2},{\hat{\alpha }}_{3}\right)$ for the log-logistic and log-Gompertz models.

To select the best model for the data of each period of the pandemic, we compare the three log-growth models by using as a criterion the mean square error (MSE) and the model selection criteria AIC and BIC, which are calculated according to the following expressions
$$MSE=\frac{1}{n}\sum_{t=1}^{n}{\left({\hat{Y}}_{t}-Y_{t}\right)}^{2},\;\;AIC=-2l\left(\hat{\theta }|y\right)+2p\;\;\mathrm{and}\;\;BIC=-2l\left(\hat{\theta }|y\right)+plog\left(n\right),$$
where ${\hat{Y}}_{t}$ is the estimated value by the model, $l\left(\hat{\theta }|y\right)=log(L\left(\hat{\theta }|y\right)$, in which, $L(\hat{\theta }|y)$ and *p* are the maximum value of the likelihood function and the number of parameters in the model, respectively. The best model is the one that has the smallest MSE, AIC and BIC values. At this point, one could opt to use another criterion of model choice, such as, for instance, the likelihood ratio test (LRT). However, as discussed by [18], there are two complications in the use of LRT for non-nested models: (1) the asymptotic distribution of the LRT under the null hypothesis will not, in general, be chi-squared, and can be difficult to evaluate mathematically; and (2) it is not clear whether model A or model B should be treated as the null model when performing the hypothesis test. In addition, if one assumes that the LRT is chosen from the three considered models, we need to make three tests (exponential × logistic), (exponential × Gompertz), and (logistic × Gompertz). Due to these two issues and the simplicity of obtaining the values of AIC and BIC using the R software, this led us to consider them as model selection criteria instead of the LRT.

Without loss of generality and for the facility of presentation, we opt to describe how to determine the change points $d_{j}$ in the next section, which describes the updating of the model as the pandemic evolved in the State of São Paulo.

## 3. Results

In this section, we present the modeling procedure adopted in the course of 411 days (from 17 March 2020 to 30 April 2021) of the pandemic in the state of São Paulo, Brazil. The first analysis was done thirty days after the record of the first death, and then, every thirty days, the model was updated.

Our choice for updating the model to every thirty days is based on the fact that, on average, the first symptoms of COVID-19 manifest between 4 to 14 days; and that the patients with grave symptoms remain an average time of 21 days in intensive clinical care. Then, it is very plausible that the changes in the evolution in the number of deaths occur between 25 to 35 days. Due to this, we opt to update the model every 30 days. However, this does not restrict the approach and a user can update the model by using another period of time, such as 20, 10 or 5 days.

### 3.1. Fitting of a Single Growth Model

For the first analysis, consider the cumulative number of deaths recorded in the first thirty days since the recording of the first death, i.e., the period from t=0 (17 March 2020) to t=29 (15 April 2020). Let $D_{1}=\left\{y_{0},\ldots ,y_{29}\right\}$ be the cumulative number of deaths log-transformed.

Once the dataset $D_{1}$ is defined, we fit the three log-growth models described in Table 1. Table 2 shows the MSE, AIC and BIC values for the three fitted log-growth models. The smallest values are highlighted in bold. As one can note, the log-Gompertz model is the best model.

Table 3 shows the point estimates and the confidence intervals (95%) for the parameters of the log-Gompertz model. The fitted model is given by
$${\hat{Y}}_{t}^{\left(1\right)}=7.0025-6.97\exp\left\{-0.0861t\right\},$$
for t≥0, where ${\hat{Y}}_{t}^{\left(1\right)}$ denotes the fitted model in the first analysis.

Figure 3a shows the values of the dataset $D_{1}$ and the fitted model (black line) for a period of 60 days (from day 0 to day 59), with 30 days of fitting and 30 days of projections. Figure 3b shows the graphic in the original scale. According to the point-estimates presented in Table 3, the projection for the maximum number of deaths is 1,099 (rounded exp{7.0024} value) and the ordinate of the inflection point is $T=\frac{log(6.97)}{0.0861}=22.5507$, i.e., the peak of the pandemic was projected to occur on the 23rd day (08 April 2020) with 484 deaths. Until the 23rd day, 496 deaths were registered. That is, the projected value for the 23rd day presented an absolute percentage error of 2.42% in relation to the recorded value. Although the projections do not indicate a very critical future situation, these results should be interpreted with great caution, since the recorded values on these thirty days represent only the beginning of the pandemic.

Unfortunately, the recorded values in the next thirty days (from 16 April 2020 to 15 May 2020) did not follow the projections of the initial fitted model since the recorded values were all above the projections, as shown in Figure 4. Due to this, we insert these thirty cumulative values (log-transformed) into the dataset $D_{1}$ and obtain the dataset $D_{1}^{u}=\left\{D_{1}\right\}\cup \left\{y_{30},\ldots ,y_{59}\right\}$, which we refer to as updated $D_{1}$. Then, we update the model considering the dataset $D_{1}^{u}$.

Once the dataset $D_{1}^{u}$ is defined, we repeat the fitting procedure of the three log-growth models. Table 4 shows the MSE, AIC and BIC values for the three fitted log-growth models. The smallest values are highlighted in bold. Again, the log-Gompertz model is the best model.

Table 5 shows the point estimates and the confidence intervals (95%) for the parameters of the log-Gompertz model. The fitted model is given by
$${\hat{Y}}_{t}^{\left(2\right)}=8.5646-7.4705\exp\left\{-0.05t\right\},$$
for t>0, where ${\hat{Y}}_{t}^{\left(2\right)}$ denotes the fitted model in the second analysis.

Figure 5 shows the values of $D_{1}^{u}$ and the curves of the models fitted on the 30th and 60th day, respectively, in the log-scale and in the original scale for a period of 90 days, with sixty days of fitting and thirty days of projection.

The MSE of the updated model ${\hat{Y}}_{t}^{\left(2\right)}$ is 0.0473, for t≥0. That is, an MSE value greater than that of the first fitted model. Besides, as one can note, the last ten recorded values are far from the projected values by the updated model and residuals are all positive, making them serially correlated and, hence, violating the initial assumption that residuals are uncorrelated. This result made us conjecture that the fit of a single growth model may not be adequate since a new period with a growth rate and/or growth type different from the first 30 days could be beginning.

In order to empirically verify our conjecture, we registered the cumulative values in the next thirty days after the date 15 May 2020 (from 16 May 2020 to 14 June 2020) and plot these values as shown in Figure 6. As one can note, our conjecture is very plausible. Due to this, hereafter, we adopt the fit of a piecewise growth model.

### 3.2. Fitting of a Piecewise Growth Model

Consider $D=\{y_{0},\ldots ,y_{89}\}$ as the cumulative number of deaths, log-transformed, recorded in the first ninety days since the first case. Let $D_{1}$ and $D_{2}$ be two sub-datasets of D representing two distinct periods of the pandemic. In order to define the change point *d* and to obtain the sub-datasets $D_{1}$ and $D_{2}$, we adopt the following approach. Let G={18,…,58} be a grid from 18 to 58 with increments of size 1. Each increment of one unit represents the increase of one day in the pandemic. Then, we define the following 41 scenarios: $D_{1}=\left\{y_{0},\ldots ,y_{d}\right\}$ and $D_{2}=\left\{y_{d+1},\ldots ,y_{89}\right\}$, for d∈G.

For each of the scenarios, we fit the three growth models to sub-datasets $D_{1}$ and $D_{2}$. We select the best model for each of the sub-datasets by using the MSE, AIC and BIC as a criterion. In addition, the best *d* value is the one associated with the fitted model that has the lowest MSE value.

According to the criteria considered, the best model is composed of a log-Gompertz model for $D_{1}$ and $D_{2}$. Figure 7 shows the MSE values of the fitted piecewise model for each *d* value considered, d∈G. The point d=20 has lead to a piecewise model with the smallest MSE value. Due to this, we set up d=20 as the separation point for $D_{1}$ and $D_{2}$.

Thus, let $D_{1}=\left\{y_{0},\ldots ,y_{20}\right\}$ and $D_{2}=\left\{y_{21},\ldots ,y_{89}\right\}$. Table 6 shows the MSE, AIC and BIC values for the three fitted growth models for $D_{1}$ and $D_{2}$. The smallest values are highlighted in bold, indicating the log-Gompertz model for $D_{1}$ and $D_{2}$.

Table 7 shows the point estimates and the confidence intervals (95%) for parameters of the log-Gompertz model for $D_{1}$ and $D_{2}$, respectively. The fitted model is given by the following piecewise model
(3)Yt(3)=6.0242−6.3704exp{−0.1227t},for 0≤t<21;10.0677−4.3578exp{−0.0242t},for t≥21;,
where ${\hat{Y}}_{t}^{\left(3\right)}$ denotes the fitted model in the third analysis.

Figure 8 shows the values of the sub-datasets $D_{1}$ and $D_{2}$ and the fitted model ${\hat{Y}}_{t}^{\left(3\right)}$ (black line) for a period of 120 days (from day 0 to day 119), with 90 days of fitting and 30 days of projection, for t≥0. The MSE of the fitted piecewise model is 0.0067. This value is smaller than the MSE of the model fitted in Section 3.1, meaning that the piecewise model fits the data better.

At this point, it is important to note that future projections are given by the log-Gompertz model fitted for sub-dataset $D_{2}$. According to point estimates for the parameters of this model, the projection for the maximum number of cases is 23,569 (rounded exp{10.0677} value) and the ordinate of the inflection point is $T=\frac{log(4.3578)}{0.0242}=60.8251$, i.e., the peak of the pandemic was projected to occur on the 81st day (20 + 61) day (17 May 2020) with 8710 deaths. There were 8842 deaths recorded by the 81st day, an absolute percentage error of 1.49% in relation to the real value.

### 3.3. Updates of the Piecewise Model

Following the procedure described in Section 3.2, we update the piecewise model every thirty days. However, in this section, we focus on describing the modeling procedure adopted to obtain the piecewise model. That is, we focus on the periods at which the change in the pandemic’s growth behavior happened.

In the updating ran on the 120th day (fourth analysis), the log-Gompertz model for $D_{2}$ was maintained. However, on the update run on the 150th day (fifth analysis), it was possible to identify a third period of the pandemic. For the identification of this third period, we consider the following procedure. Let $D_{2}=\left\{y_{21},\ldots ,y_{d}\right\}$ and $D_{3}=\left\{y_{d+1},\ldots ,y_{149}\right\}$, for d∈G={60,…,110}, where G is a grid from 60 to 110 with increments of size 1, representing possible change points of the pandemic. Repeating the procedure of fitting the three growth models for each sub-dataset, we obtain that the log-Gompertz is the best model for both sub-datasets.

Figure 9 shows the MSE values from models fitted for d={90,…,100}. As one can note, d=92 is the best change point, i.e., the point with the smallest MSE value. Thus, we set $D_{2}=\left\{y_{21},\ldots ,y_{92}\right\}$ and $D_{3}=\left\{y_{93},\ldots ,y_{149}\right\}$.

The fitted model is given by the following piecewise model
(4)Y^t(5)=6.0242−6.3704exp{−0.1227} 0≤t<21;10.0707−4.3603exp{−0.0241t},for 21≤t<92;10.8892−1.5467exp{−0.0132t},for  t≥92;,
where ${\hat{Y}}_{t}^{\left(5\right)}$ denotes the fitted model in the 5-th analysis.

Figure 10 shows the values of the sub-datasets $D_{1}$, $D_{2}$ and $D_{3}$ and the fitted model ${\hat{Y}}_{t}^{\left(5\right)}$ (black line) for a period of 180 days (from day 0 to day 179), with 150 days of fitting and 30 days of projection. The MSE of the fitted piecewise model is 0.0004811.

Following this procedure, after 14 analysis steps, we identify seven different periods of the pandemic with change points on the days d={20,92,240,295,363,381}. The piecewise growth model has the following configuration: log-Gompertz for $D_{1}=\left\{y_{0},\ldots ,y_{20}\right\}$, $D_{2}=\left\{y_{21},\ldots ,y_{92}\right\}$, $D_{3}=\left\{y_{93},\ldots ,y_{240}\right\}$, log-exponential for $D_{4}=\left\{y_{241},\ldots ,y_{295}\right\}$, $D_{5}=\left\{y_{296},\ldots ,y_{363}\right\}$, $D_{6}=\left\{y_{364},\ldots ,y_{381}\right\}$ and log-Gompertz for $D_{7}=\left\{y_{382,\ldots }\right\}$. Its mathematical expression is given by
(5)Y^t(14)=6.0242−6.3704exp{−0.1227t},for  0≤t<21;10.1376−4.4034exp{−0.0229t},for 21≤t<93;10.7378−3.1148exp{−0.0158t},for  93≤t<241;10.5955+0.0030t,for  241≤t<296;10.7666+0.0043t,for  296≤t<364;11.0592+0.0091t,for  364≤t<382;11.7423−0.5207exp{−0.0226t},for  t≥382;,
where ${\hat{Y}}_{t}^{\left(14\right)}$ denotes the fitted model in the 14-th analysis step.

Figure 11 shows the recorded values and the curve of the fitted piecewise model ${\hat{Y}}_{t}^{\left(14\right)}$, in the log and original scale, for a period of 600 days. The MSE of the fitted model is 0.0042, t≥0. At this point of the analysis, the future projection is given by the log-Gompertz model fitted for sub-dataset $D_{7}$. According to the point estimated for the parameters of this model, the projection for the maximum number of deaths is 122,296 (exp{11.7142}). In other words, if the current scenario is maintained, it is expected that there will be additional 26,105 deaths.

Table 8 shows the projections of the model for the cumulative number of deaths in the last ten days of the sub-dataset $D_{7}$ and the recorded values. The fourth line of this Table shows the absolute percentage error in relation to the real value. The biggest percentage error was 0.7877%.

In addition to modeling the different pandemic periods, the fitted piecewise model also allows linking the model change points to facts that affect the speed of the epidemic spreading. For example, on the 334th day (13 February 2021) the first case of COVID-19 by the $P_{1}$ variant was registered in the state of São Paulo. After thirty-one days, on the 364th day (15 March 2021), the proposed approach has indicated a change point; changing from an exponential model with a growth rate of 0.0043 to an exponential model with a growth rate of 0.0091. That is, a change for a more aggressive pandemic phase occurred. Since the disease has an average time from 4 to 14 days to manifest the first symptoms and the patients with grave symptoms remain an average time of 21 days in intensive clinical care; then, it is very plausible to consider that the $P_{1}$ variant is one of the possible factors related to the change point.

On the other hand, due to an excessive increase in the number of deaths due to the COVID-19 in the first months of 2021, the government of the state of São Paulo has published on 6 March 2021 (45-th day) a decree implementing an emergency phase to contain the transmission of the disease. In this decree, a curfew from 8 pm to 5 am and the closing of commerce was adopted. Twenty-eight days after the decree, the proposed approach has identified a change from the exponential model to a Gompertz model (day 382—4 February 2021). That is, a change to a “little better” phase occurred because, in this new phase, at least, it is possible to make a projection for the occurrence of a peak. Although the modeling procedure does not allow us to associate directly the publication of the decree with the “new” pandemic phase, it is very plausible to consider that the adoption of the decree has contributed to this change.

To finalize this section, we inform the reader that the fitted piecewise model in Equation (Equation 5) is not absolutely continuous. However, in the opposite to the fit of a single growth model, the absolute continuity of a piecewise model is not a fundamental property because the aim is to identify change points at which model changes happen. Thus, the final model may have a jump in the change points. However, as is expected for a growth model, the piecewise fitted model has the following properties:f(t) is non-decreasing;f(t) is continuos to the right side of the change points;$\underset{t\to -\infty }{lim}f\left(t\right)=0$ and $\underset{t\to +\infty }{lim}f\left(t\right)=\alpha_{1}^{*}$, where $\alpha_{1}^{*}$ is the upper asymptote of the last function.

### 3.4. Piecewise Growth Model for the Cumulative Number of Cases

As another illustration of the good performance of the proposed method, we fit a piecewise model for the cumulative number of cases recorded in the State of São Paulo, Brazil, in the period from 26 February 2020 to 30 April 2021. In this period, 2,903,709 cases of COVID-19 were recorded.

The cumulative number of cases also presented a heterogeneous behavior over time. In the course of analysis, we identify eight different periods for the number of cases with change points on the days d={34,93,144,256,310,346,376}. However, at this point, it is important to emphasize that the number of cases is directly connected with the number of diagnostic tests performed. Thus, when a change in the pandemic’s behavior is detected, this may be due to increased testing.

The fitted piecewise model has the following mathematical expression
(6)Z^t(14)=7.5950−log1+4214.0455exp{0.2850},for  0≤t<35;13.1173−5.2413exp{−0.0192t},for  35≤t<94;14.3067−2.8624exp{−0.0143t},for  94≤t<145;14.0132−1.1061exp{−0.0228t},for 145≤t<257;13.9311+0.0048t,for  257≥t<311;14.1852+0.0067t,for  311≥t<347;14.4121+0.0047t,for  347≥t<377;15.7549+1.1918exp{−0.0061t},for  t≥377;,
where ${\hat{Z}}_{t}^{\left(14\right)}$ denotes the fitted model for the number of cases in the 14-th analysis.

Figure 12 shows the recorded values and the curve of the fitted piecewise model ${\hat{Z}}_{t}^{\left(14\right)}$, in the original scale, for a period of 1000 days. The MSE of the fitted model is 0.0042. At this point of the analysis, the future projections are given by the log-Gompertz model fitted for sub-dataset $D_{8}$. According to point estimates for parameters of this model, the projection for the maximum number of cases is 6,954,504 (exp{15.7549}).

## 4. Final Remarks

In this paper, we describe a case study on the evolution of the COVID-19 pandemic in the state of São Paulo, Brazil. The main aim of the study was to fit a piecewise growth model in order to be able to explain the different periods of the pandemic’s evolution. In addition, there is also the interest in being able to give a short-term forecast for the cumulative number of cases and predict the peak point of the pandemic for each period of the pandemic.

The modeling procedure was developed in the course of time by updating the model every thirty days. In addition, for each update, we also verify the existence of two different periods of the pandemic. In the affirmative case, we separate the dataset of the period into two sub-datasets and fit a growth model for each of the sub-datasets. Overall, we identify seven periods of the pandemic for the cumulative number of deaths. To each period of the pandemic, we fit three non-linear growth models: exponential, logistic and Gompertz. In order to select the best model for each sub-dataset, we consider the MSE, the AIC and BIC as criteria.

The fitted piecewise model for the number of deaths is a mix of the Gompertz model and the exponential model. From a practical viewpoint, this can be viewed as the main advantage of the proposed method, because, it is capable of explaining each period by different values of pandemic parameters, such as the peak and the growth rate for each period. In addition, the results also show that the fit of a piecewise model is more effective than the fit of a single growth model.

From a practical viewpoint, the identification of the change points shows for government agencies that some containment procedures of the transmission of the disease need to be implemented; or, if some containment procedures implemented are working. For example, if the data from a period are explained by a Gompertz model, then there is a projection for the peak of the pandemic (inflection point) and an estimate for the maximum number of cases (upper asymptote). Under this scenery, the government’s agents may elaborate better strategies for containment of the transmission of the virus. However, if a change point is identified, in which, the recorded data after this change point are explained by an exponential model, then there is a new situation without a projection for a peak. That is, the situation changed to a more aggressive phase of the pandemic. This scenery shows that that more restrictive containment strategies need to be implemented as soon as possible.

On the other hand, if we identify a change point from a period explained by an exponential model for a period explained by a Gompertz model; this indicates that a change happened towards “a little better” situation because now, at least, it is possible to project the peak of the pandemic. This information may be used by government agencies in order to elaborate and justify the adoption of containment strategies of the transmission of the disease, and, in this way, to obtain a flattening of the Gompertz curve and, consequently, anticipate the occurrence of the peak and reduce the number of deaths (decrease the upper asymptote value).

Both examples described above show that it is important to consider the fit of a piecewise growth model, such as the one in our proposal, in contrast to fitting a single growth model. Although the paper has been developed considering only three growth models (exponential, logistic, and Gompertz), the procedure presented can be easily adapted for other kinds of growth models. The computational codes used for fitting the models are in the R language and can be requested by contacting the authors via e-mail.

## Figures and Tables

**Figure 1 entropy-23-01013-f001:**
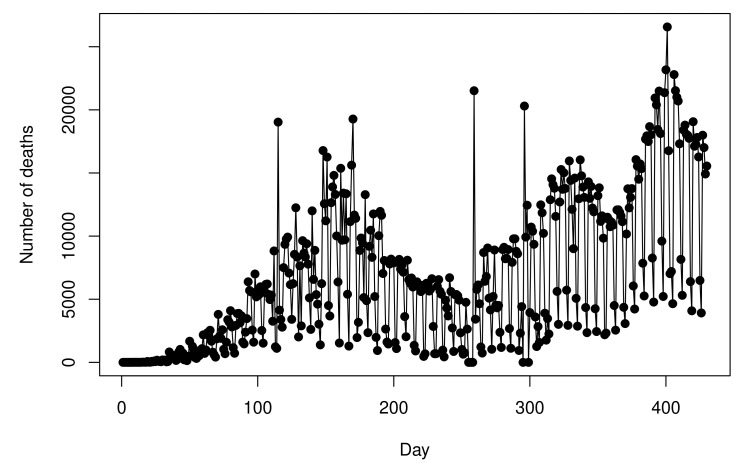
Number of deaths recorded by day.

**Figure 2 entropy-23-01013-f002:**
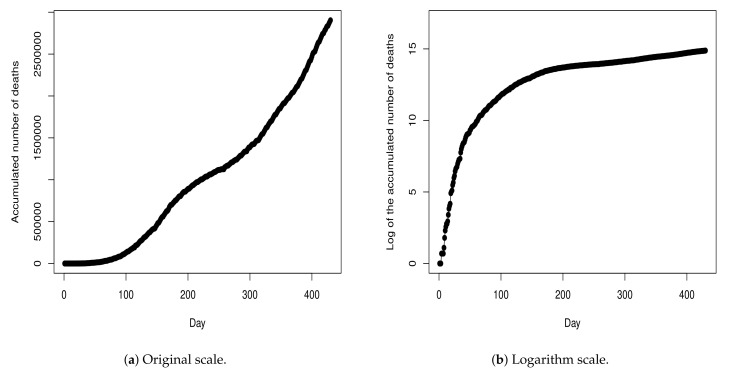
Cumulative number of deaths until day *t*, for t=0,1,…,n.

**Figure 3 entropy-23-01013-f003:**
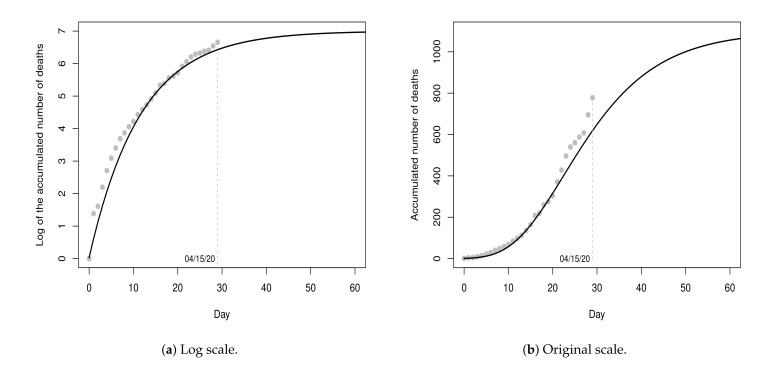
Data from 17 March 2020 to 15 April 2020 and fitted model ${\hat{Y}}_{t}^{\left(1\right)}$, for t≥0.

**Figure 4 entropy-23-01013-f004:**
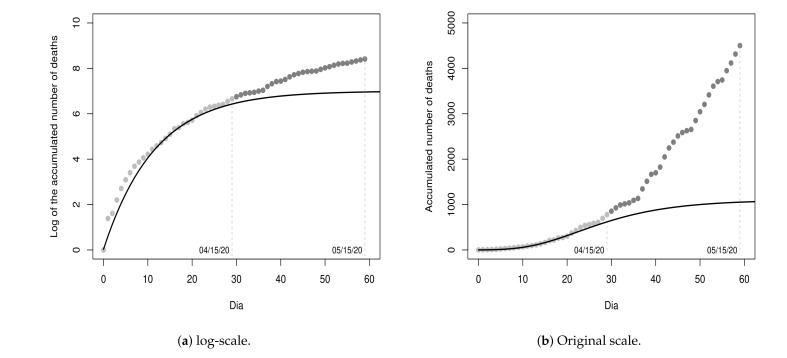
Data from 17 March 2020 to 15 May 2020 and fitted model ${\hat{Y}}_{t}^{\left(1\right)}$, for t≥0.

**Figure 5 entropy-23-01013-f005:**
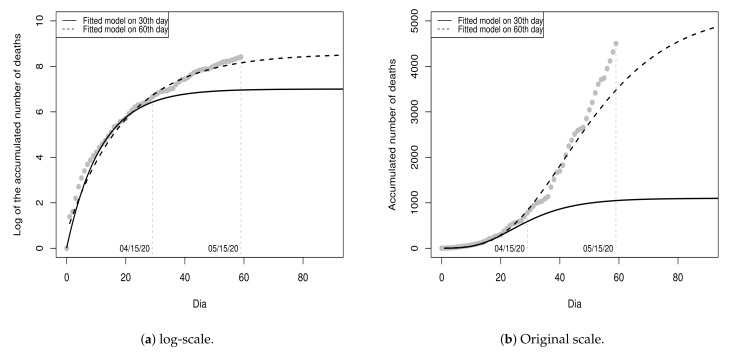
Data from 17 March 2020 to 15 May 2020 and fitted models on 30th $\left({\hat{Y}}_{t}^{\left(1\right)}\right)$ and 60th day $\left({\hat{Y}}_{t}^{\left(2\right)}\right)$.

**Figure 6 entropy-23-01013-f006:**
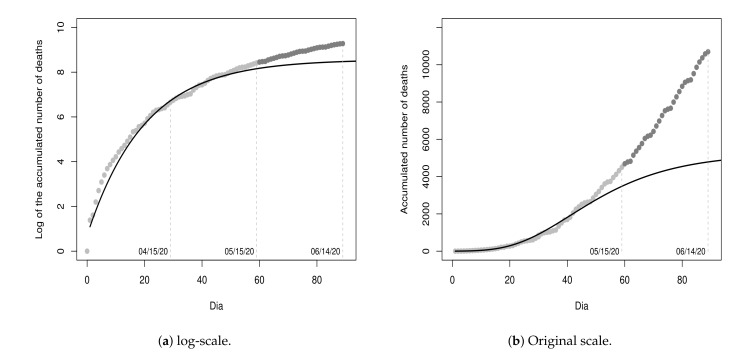
Data from 17 March 2020 to 14 June 2020 and fitted models on 30th $\left({\hat{Y}}_{t}^{\left(1\right)}\right)$ and 60th day $\left({\hat{Y}}_{t}^{\left(2\right)}\right)$.

**Figure 7 entropy-23-01013-f007:**
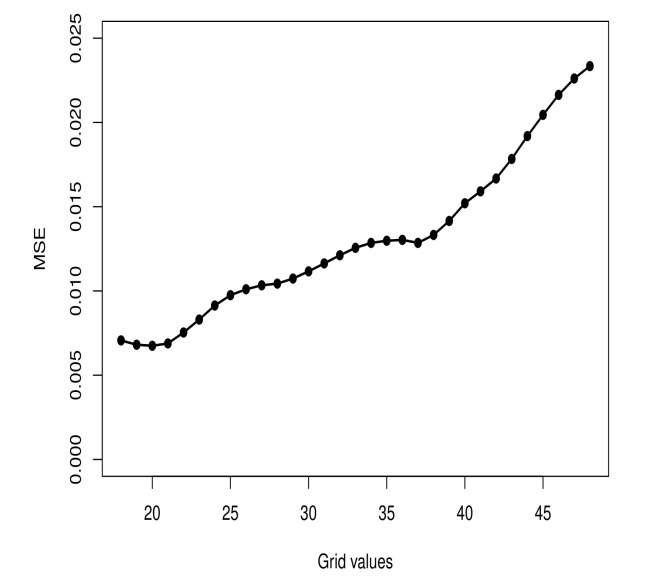
MSE values for fitted models according to the grid G.

**Figure 8 entropy-23-01013-f008:**
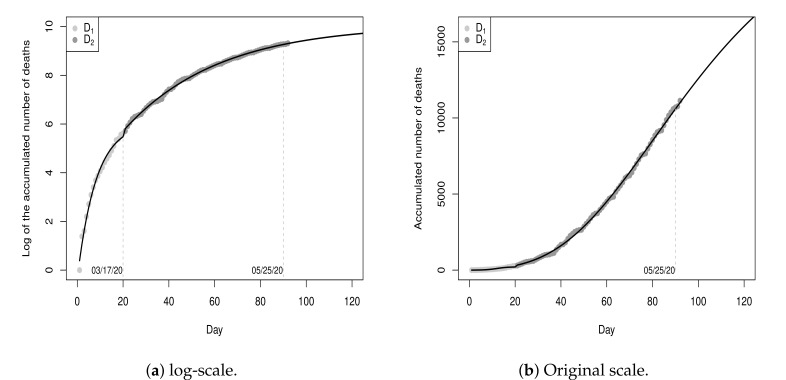
Data from 26 February 2020 to 25 May 20 and fitted piecewise model ${\hat{Y}}_{t}^{\left(3\right)}$, t≥0.

**Figure 9 entropy-23-01013-f009:**
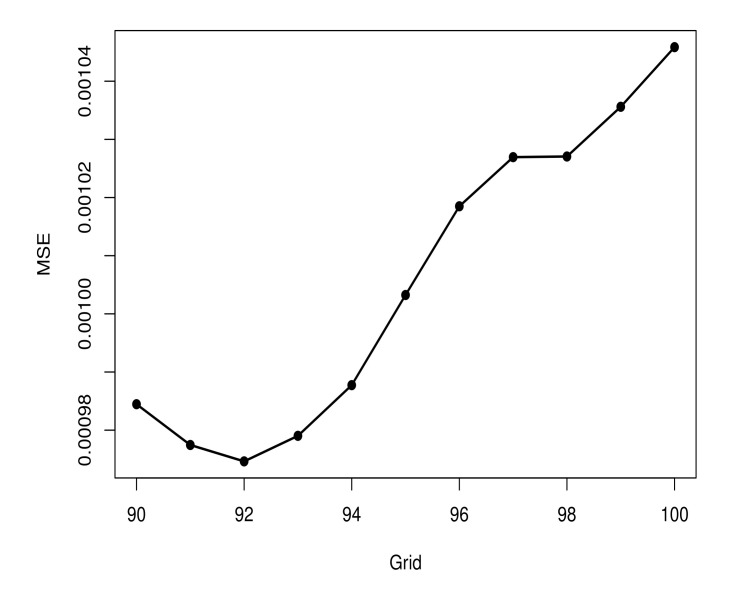
MSE values for fitted models according to the change point *d*.

**Figure 10 entropy-23-01013-f010:**
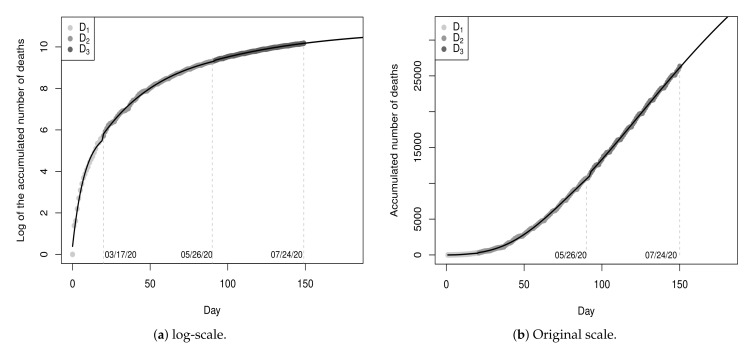
Data from 26 February 2020 to 25 May 20 and fitted piecewise model ${\hat{Y}}_{t}^{\left(5\right)}$, t≥0.

**Figure 11 entropy-23-01013-f011:**
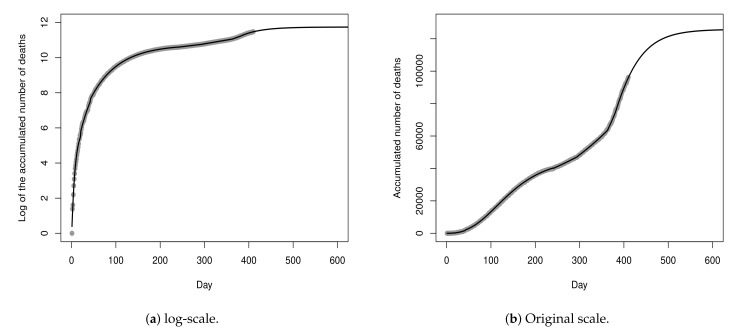
Data from 26 February 2020 to 30 April 2021 and fitted piecewise model ${\hat{Y}}_{t}^{\left(14\right)}$, t≥0.

**Figure 12 entropy-23-01013-f012:**
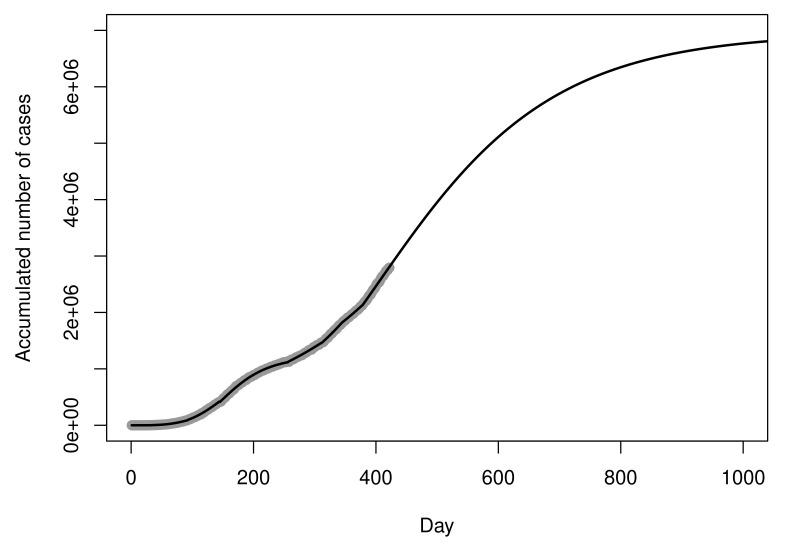
Fitted piecewise model for the cumulative number of cases.

**Table 1 entropy-23-01013-t001:** Log-transformed growth models.

Model	Parameters	Mathematical Expression	Inflection Point	Growth at Inflection Point
log-exponential	$\theta =(\alpha_{1},\alpha_{2})$	$Y_{t}=\alpha_{1}+\alpha_{2}t$	does not exist	does not exist
log-logistic	$\theta =(\alpha_{1},\alpha_{2},\alpha_{3})$	$Y_{t}=\alpha_{1}-log\left(1+\alpha_{2}\exp\left\{-\alpha_{3}t\right\}\right)$	$T=log\left(\alpha_{2}\right)/\alpha_{3}$	$f\left(T|\theta \right)=\alpha_{1}-log\left(2\right)$
log-Gompertz	$\theta =(\alpha_{1},\alpha_{2},\alpha_{3})$	$Y_{t}=\alpha_{1}-\alpha_{2}\exp\left\{-\alpha_{3}t\right\}$	$T=log\left(\alpha_{2}\right)/\alpha_{3}$	$f\left(T|\theta \right)=\alpha_{1}-1$

**Table 2 entropy-23-01013-t002:** MSE, AIC and BIC values for the fitted models for dataset $D_{1}$.

Model	Criterion		
	MSE	AIC	BIC
log-exponential	0.2776	52.6845	56.8881
log-logistic	0.0968	23.0901	28.6949
log-Gompertz	**0.0306**	**−11.5014**	**−5.8966**

**Table 3 entropy-23-01013-t003:** Parameter estimates for model $Y_{t}^{\left(1\right)}$.

Values	Parameter		
	$\alpha_{1}$	$\alpha_{2}$	$\alpha_{3}$
Estimates	7.0025	6.9700	0.0861
C. interval	(6.6901, 7.3981)	(6.6779, 7.2712)	(0.0734, 0.0994)

**Table 4 entropy-23-01013-t004:** MSE, AIC and BIC values for the fitted models for dataset $D_{1}^{u}$.

Model	Criterion		
	MSE	AIC	BIC
log-exponential	0.4724	131.2807	137.5637
log-logistic	0.1700	71.9576	80.3350
log-Gompertz	**0.0473**	**−4.8170**	**3.5604**

**Table 5 entropy-23-01013-t005:** Parameter estimates for model $Y_{t}^{\left(2\right)}$.

Values	Parameter		
	$\alpha_{1}$	$\alpha_{2}$	$\alpha_{3}$
Estimates	8.5646	7.4705	0.0500
C. interval	(8.3497, 8.8117)	(7.2472, 7.6960)	(0.0452, 0.0551)

**Table 6 entropy-23-01013-t006:** MSE, AIC and BIC values for the models fitted for datasets $D_{1}$ and $D_{2}$.

Model	Dataset $D_{1}$			Dataset $D_{2}$		
	MSE	AIC	BIC	MSE	AIC	BIC
log-exponential	0.2012	30.6927	33.6799	0.0456	−11.5210	−4.7755
log-logistic	0.0716	12.0195	16.0024	0.0046	−169.9950	−161.0010
log-Gompertz	**0.0242**	**−9.6880**	**−5.7051**	**0.0017**	**−237.1040**	**−228.1100**

**Table 7 entropy-23-01013-t007:** Estimates for the piecewise model parameters $Y_{t}^{\left(3\right)}$.

Parameter	Dataset	
	$D_{1}$	$D_{2}$
$\alpha_{1}$	6.0242	10.0677
	(5.6457, 6.5511)	(9.9692, 10.1756)
$\alpha_{2}$	6.3704	4.3578
	(6.0233, 6.7403)	(4.2767, 4.4464)
$\alpha_{3}$	0.1227	0.0242
	(0.0984, 0.1484)	(0.0230, 0.0254)

**Table 8 entropy-23-01013-t008:** Projections and percentage error for the period from 11/09/20 to 11/16/20.

Date	04/13/21	04/14/21	04/15/21	04/16/21	04/17/21	04/18/21	04/19/21	04/20/21	04/21/21	04/22/21
Projection	90,279	90,949	91,609	92,259	92,899	93,529	94,149	94,759	95,359	95,950
Real value	90,627	90,810	91,673	92,548	92,693	92,798	93,842	94,656	95,532	96,191
error	348	139	64	289	206	731	307	103	173	241
% error	0.3840	0.1531	0.0698	0.3123	0.2222	0.7877	0.3271	0.1088	0.1811	0.2505

## Data Availability

https://www.seade.gov.br/coronavirus (accessed on 13 July 2021).

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
