# Peer review of "Piecewise Modeling the Accumulated Daily Growth of COVID-19 Deaths: The Case of the State of São Paulo, Brazil"

_entropy, 2021, doi:10.3390/e23081013_

Round 1

Reviewer 1 Report

The authors present a statistical approach for modeling the time evolution of cumulated COVID-19 deaths in Sao Paulo, Brazil, based on piecewise fitting of different types of statistical log-growth models. The approach taken differs from most other contemporary studies in that it rejects any mechanistic assumptions on the processes leading to the empirically observed numbers. While this could provide complementary insights to existing process-based epidemiological models, the authors have failed to convince me of (i) the formal correctness and (ii) the usefulness of their approach, for several reasons that I will elaborate on below.

Presentation quality: While the text is generally understandable, the number of (mostly minor) grammatical errors is rather larger and calls for careful proofreading. For example, consistently avoiding the correct use of the plural form of “deaths” is a bit annoying when reading it several times per page. Some of the tables listing just the parameter estimates (plus confidence intervals) for specific models appear unnecessary and should be replaced by inline information. The overall presentation appears a bit careless for particularly two reasons: (i) the entire text of the introduction is duplicate (ll.56-111 equal ll.2-56), which should not appear within a careful submission; (ii) the authors refer to Supplementary Materials (external to the manuscript), cf. ll.55-56, while providing the corresponding information as Appendices (inside the manuscript). The numbering of the appendices is odd and should have been realized by the authors before submission. While background on the employed statistical methods is provided to a sufficient extent, background on the study “subject” COVID-19 is almost completely missing (surprisingly given the exhaustive amount of literature on this topic that appeared in the last year). An appropriate embedding of the present work into the current state of research is more or less completely missing. Likewise, a discussion of the obtained results (putting them into the broader context) is entirely missing.

Major comments:

  1. The selection of “periods” (respectively, the approach to define them) appears not well justified. Especially, the “change points” for switching from one statistical model to another are not discussed at all in terms of epidemic stages, intervention measures, or any other available knowledge on the evolution of the pandemic in Sao Paulo. Without such discussion, the obtained results present a mere statistical exercise (which could be interesting for a statistics community if the methodological approaches would be new and correctly employed, but appear of limited value in the context of COVID 19 research).
  2. In general, I feel the motivation of the statistical model used very ad hoc and disregarding any process knowledge on epidemic spreading. The different log-growth models have been derived theoretically under certain assumptions (e.g., an infinite reservoir in case of exponential growth, which can be easily discarded for an epidemic process). The motivation for including exponential, logistic and Gompertz functions in the presented analysis is rather poor and should be substantiated by mechanistic reasoning.
  3. You can fit any piecewise model, but then you should explain why the parameters (and possibly model type) can change abruptly during the pandemic. The meaning of this is not clear at all.
  4. In Section 3.2, when introducing the two-interval model, the optimal breakpoint is found at the edge of the interval of possible breakpoints that is studied (Figure 7). This may indicate that the statistically optimal solution might rather be provided by a breakpoint that occurs even earlier than day 20.
  5. Using the fitted model for extrapolating the evolution of the death toll of COVID 19 for two more years into the future (Fig. 12) appears pretty unjustified given intervention measures (vaccination campaigns), emergence of mutations, and other relevant aspects that control the corresponding epidemiological process.
  6. While I may accept representing the deaths in logarithmic units (and hence attempting to fit log-growth models), a very basic property of any type of least square regression is that it does not provide unbiased estimates of the actual (non-logarithmic) process parameters, since minimizing sums of squared differences between linear and logarithmic quantities is not equivalent. So even if I would understand the rationale of providing a proper statistical model for the growth of cumulative deaths, the model parameters (and even the selected model structure) might not be optimal for the property of interest if optimized for its logarithm only.

Specific comments:

  • Abstract, first line: COVID-19 is the disease caused by the new corona virus, not the name of the virus (which is rather SARS-CoV-2). When attempting to publish on this subject, at least this elementary difference should be realized by the authors.
  • 1, l.6: “This is the Brazilian case!” does not sound like a very scientific statement that could be expected in a scholarly paper.
  • 2, l.21: Reference [2] does not seem to have anything to do with the coronavirus context in which it is cited here.
  • 2, l.42: It is not clear why you mention those countries as motivation for the present work.
  • 2, ll.43-44: It is a bit confusing that you state before that you “only” model cumulative deaths while then also modeling cumulative cases.
  • 3, l.118 (and many further occurrences): Figure 1 shows the recorded D_t values (not “the graphic of” – this is awkward wording, especially when repeating it with every new figure)
  • 6, ll.146-147: different citation style than in the rest of the paper
  • 7: The model equation and parameters from Table 3 should be simply given as text; both are neither interesting nor relevant enough to warrant the amount of space taken by a separate equation and table. Same for Table 5.
  • 8, ll.177-181: It sound to me pretty pointless to report about an inflection point within the fitting range of just the first 30 days.
  • 7: Labelling the x axis with “grid” is completely pointless.
  • 18, l.284: “able to explain the different periods” – I do not see any explanation throughout the paper
  • Appendix 1.2: The authors should realize that the three model parameters are partially affected by one important characteristic (the reservoir size from which susceptible individuals can be taken; in an ecological context, this would be related to the “carrying capacity” of the humans+virus “ecosystem”).

Author Response

We thanks this reviewer for your comments and suggestions.
Please, find attached the point-to-point answers to the raised comments.  

Reviewer 2 Report

Comments on Entropy manuscript:  “Piecewise modeling the accumulated daily growth of COVID-19 death:  The case of the State of São Paulo, Brazil”

General comments:

  1. Many thanks for an interesting manuscript with interesting results.
  2. Could the authors add brief discussion comparing their piecewise approach to, say, loess or other smoothing approaches that fit a smooth curve to the data. The added feature of a population growth model with interpretable parameters seems to be one advantage, which would seem to better support forward projections of outcomes, as well.
  3. The manuscript would benefit from careful proofreading for subtleties in the English language. “growth of death” vs. “growth of deaths”, for example.  Some suggestions for the abstract appear below but the manuscript needs some polishing throughout.

Specific comments

  1. Title: Should “Modeling” be capitalized?
  2. Line 1 of abstract. “the new coronavirus called COVID-19” should be “the new coronavirus called SARS-CoV-2”.   The virus is SARS-CoV-2, the disease caused by SARS-CoV-2 is COVID-19.  Also, “did increase the interest in” to “increased interest in”.
  3. Line 2 of abstract. “(and death)” to “(and associated deaths)”.
  4. Line 5 of abstract. “(and death) of the COVID-19” to “(and associated deaths) due to COVID-19”.
  5. Line 6 of abstract. “the it is important that the modeling prodedure be” to “it is important that modeling procedures be”.
  6. Line 3. “as it created” to “and created”.
  7. Line 6. “leading many countries to an increase in poverty in their population and the loss of thousands of human lives” to “leading to an increase in poverty and the loss of thousands of lives in many countries.”
  8. Line 15. “these values do not depend on the number of diagnostic tests performed.” This is true but the accuracy of the COVID-associated deaths also relies on accurate reporting of COVID-19 in mortality statistics.   Is this generally the case in Brazil?  In many countries, complete testing to include in mortality statistics did not occur until several months into the pandemic, and there remain questions as to the completeness of reporting of COVID.   Can the authors mention this briefly?
  9. Lines 16-17. “without the maskering due to the increase in the number” to “without having to adjust for changes in the number”.
  10. Line 18. “The modeling” to “Our proposed modeling”.
  11. Line 28. “increments of size 1”. Do these refer to temporal increments?   What are the units?
  12. Line 37. “have got” to “fine”
  13. Line 38. “for the peak of the pandemic” to “for the peak of the pandemic in the current period.”
  14. Line 41. “closed in Population” to “close in population size”.
  15. Line 45. “the proposed modeling are” to “the proposed modeling approach are”.
  16. Lines 57-111. Delete, these are a repeat of the opening section.
  17. Line 118. “shows the graphic of the” to “shows the”. Also, do some of the low values represent weekends in the time period.  In many data sets, data were not input on weekends and regularly show low values that reflect reporting effort, not patterns in mortality.  If this is the case in São Paulo, it would be helpful to mention here.
  18. Line 136. The models assume no temporal correlation, could this be added to the model, would it be helpful?
  19. Line 138. “that present” to “that presents”. Also, “indefinitely growths” to “indefinite growth”.
  20. Line 139. “Gompertz has a growth limit” to “Gompertz have growth limits”.
  21. Table 1. “Grows at inflection point”. Should “grows” be “growth”?
  22. Line 156. “adequated” to “adequate”.
  23. Line 158. “loss o generality” to “loss of generality”.
  24. Line 159-160. “in which is described” to “which describes”.
  25. Line 186. “of the fitted model” to “of the initial fitted model”.
  26. Line 188. To help the reader, perhaps the notation can be adjusted so that there are not multiple definitions of D_1?   I understand the logic, but the notation can get confusing.
  27. Figure 7. Would it be possible to plot the MSE for grid points 18 and 19, to illustrate whether the MSE is minimized at grid point 20 or whether the MSE continues to fall for earlier grid points?
  28. Line 229. “that the future projections is” to “that future projections are”.
  29. Line 234. “Until the 81th day were recorded 8.842 death. An absolut” to “There were 8,842 deaths recorded by the 81st day, an absolute”.
  30. Figure 11 (and others). Y-axis label.  “Accumulated number of death” should be “Accumulated number of deaths.”
  31. Line 270. “have present a heterogeneous” to “presented heterogeneous”.
  32. Line 271. “in the couser of anlaysis” to “in the course of analysis.
  33. Line 298. “their pandemic parameters” to “different values of pandemic parameters”.
  34. Line 315. “more inclined is the curve” to “increase the rate of growth curve”.
  35. Line 320. “In the opposite of” to “In contrast to”.

Author Response

(The authors gave the same response as above.)

Reviewer 3 Report

In this manuscript, the authors present a modeling procedure based on the fit of a piecewise growth model for the cumulative number of deaths. The model is updated in the course of the pandemic, and when a new period of the pandemic is identified it creates a new sub-dataset composed of the cumulative number of death registered from the change point and a new growth model is chosen for that period. Three growth models were fitted for each period: Exponential, Logistic and Gompertz models. The method is illustrated by a case study for the cumulative number of deaths due to COVID-19 in Sao Paulo State, Brazil. 

In my opinion, the authors make the same mistake that could be seen in many works concerning the ongoing epidemic. Namely, they choose some functions and try to fit them to data of the epidemic. Of course, any function can be fitted to the data for a longer or shorter time interval. However,  this will not say much about the dynamics of the epidemic. This can be seen in the first fitted curves shown in the manuscript: after the fitted period, the data did not really follow the predictions. Hence, the authors claim that at such points, a new period will start and choose another function to be fitted to the data of this new period. However, this could be done with arbitrarily chosen functions as well. Retrospectively, one can obtain a fit to the data in such a way, but I cannot see if this explains anything about the underlying phenomenon. One way I can imagine such a modelling framework can be applied would be if one could identify some patterns in  how one ot other function will follow in the fitting to data points and give predictions based on these patterns. I cannot find such conclusions in the manuscript, in fact, there are very hardly any conclusions: how can this modelling approach be useful? How can it be used for prediction? What kind of information does it provide about the current evolution of an epidemic?

In view of the above, I cannot suggest the acceptance of the manuscript.

Some minor remarks:
- Accidentally the introduction was doubled.
- The English should be revised, there are several typos, minor grammatical errors (e.g. "cumulative number of deathS") and several sentences to be reformulated.
- Although in a smaller extent, but there is also an uncertainty in the number of deaths. This might be due to imperfect reporting (several death cases will appear in the reports much later). Also, there might be differences in the definition of a COVID-19 related death case.
- A 7-day moving average might provide a dataset with less uncertainty.

Author Response

(The authors gave the same response as above.)

Reviewer 4 Report

Review of the paper:

Piecewise Modeling the accumulated daily growth of COVID-19 death: The case of the State of São Paulo, Brazil

 The paper deals with statistical modeling of covid-19 growth, which in general, has presented a heterogeneous evolution over time. It is then important that the modeling procedure be capable of identifying periods with different growth rates and proposing an adequate model for each period. The paper presents a modeling procedure based on the fit of a piecewise growth model for the cumulative number of death.  Three growth models were fitted in data Exponential, Logistic and Gompertz models. The best model for the cumulative number of death recorded is the one with the smallest mean square error and the smallest AIC and BIC values. This approach is illustrated in a case study, in which, we model the number of death of the COVID-19 recorded in the State of São Paulo, Brazil. Finally, the results have shown that the fit of a piecewise model is very effective for explaining the different periods of the pandemic evolution.

The statistical modeling is well performed and the paper is very well written, explaining in details the models and the fitting procedures. I am positive for the paper to be publish, however I think, there is an issue to be addressed:

While the mathematical part is sound, a conceptual presentation is missing or it is underdeveloped. It would be useful if the authors develop an argumentation and I deeper conceptual understanding is provided, explaining what theoretical perspective supports each model. That is, besides the ‘curve fitting’ outcomes what is the meaning of fostering each of the modeling choice. This is rather a theoretical question, which is needed to be answered and support the inductive endeavor.

Author Response

(The authors gave the same response as above.)

Round 2

Reviewer 1 Report

First of all, let me apologize to the authors for the long time required for providing my review on their revised manuscript. This has not been due to quality issues with their work, but rather heavy time conflicts at my side.

While I have originally recommended rejection of this manuscript, working through the revised version along with the reviews provided by the other referees made me conclude that the presented research may still be of interest to the readership of Entropy. Yet, I feel that substantial further revision is necessary for making this an important and broadly assessable contribution.

Major comments:

  1. While the different pandemic periods are defined based on statistical reasoning, which is well possible and correct within its intrinsic limitations, I would find it quite helpful if the authors could attempt to link these periods to the general evolution of political decisions, counter-measures, appearance of mutants, etc., which may have affected the differential speed of the epidemic spreading. Providing some more thorough idea of a possible interpretation of the different identified phases would surely make the analysis much stronger and of more practical relevance.

  1. The authors use AIC and BIC for model selection. I wonder if this is strictly valid in case of non-nested model selection as used here. Or, wouldn’t a more rigorous approach like likelihood ratio testing along with bootstrapping the underlying null distribution for each specific test be a more appropriate strategy? I suggest that the authors should briefly comment on possible methodological limitations in this regard.

  1. Page 4, above Fig. 2: A lognormal distribution cannot have a mean of zero, only a mean of one (i.e., log-mean of zero). Please correct the numbers given here accordingly.

  1. I find the two symbols \epsilon and \varepsilon used for the residual in standard and logarithmic units hard to distinguish and therefore easy to confuse. Maybe you could consider replacing one of them by another symbol?

  1. 5, after l.122: Please clarify here if the nonlinear least squares regression applies to the logarithmic or the original quantities, since by the nature of LS estimators, the regression models obtained with both approaches will not be equivalent. If you use the logarithmic quantities, it might also be good to address this along with the interpretation of the models in “normal variables”. In a similar spirit, I suppose that also the MSE is evaluated for the logarithmic rather than the original variables; this should also be explicitly mentioned.

  1. In the first part of the analysis, the authors use 30-day windows for model updates. While I can easily understand the original idea of using at least 30 days for proper model selection and parameter estimation, this restriction to the update interval is less obvious to me. Why not updating every 20, 10, 5,… days?

  1. In order to more transparently cross-refer to different model variants/update stages, I recommend numbering the successive statistical models as \hat{Y}^{(0)}, \hat{Y}^{(1)}, etc., and refer in the text and the table/figure captions explicitly to the model that is currently discussed (especially the captions are hard to relate to the different models, since they are often identical).

  1. Figure 3(a) shows that the empirical values are (almost) always(!) larger than those predicted by the model, indicating a biased model with correlated residuals. This is not what I would expect from a proper nonlinear LS estimate, so I have to ask the authors for clarification and, if applicable, correction. Note that this does not only affect the model extrapolation beyond the calibration period (that is emphasized in l.151 and following). Figure 6(a) shows essentially the same behavior, indicating again some kind of model misspecification. I have the feeling that the answer to this observation may be related to my previous comment #5.

  1. A general question regarding the three considered statistical growth models: To me, they make sense as physical process models if a population size out of which the dead people originate (i.e., the infected population) remains constant with time. However, this is obviously not the case here. I understand that the three models have been selected in a somewhat ad-hoc way to meet statistical rather than physical criteria (which may be fine), but the conceptual limitations of understanding them as mechanistic models in the present context should be discussed at least briefly.

  1. I don’t quite understand from the material presented if the piecewise fitting with different component functions implies continuity of the total curve – I don’t see this to be the case from the mathematical descriptions, yet would expect continuity to be a fundamental property of an appropriate model.

Further technical suggestions:

  • 1, l.10: “because, other than for the number of cases”
  • 1, l.25: “thousands of lives in many countries” – why not giving here an overall number of global cases at some specific data
  • 1, l.30: word order: “many articles were published”
  • 2, l.55: “a possible change point”
  • 2, l.57: “for two sub-datasets obtained according to each possible change point”
  • 2, l.67: “in which we model”
  • 2, l.75: What do you mean by “press vehicles”?
  • 2, l.75: “that these published data”
  • 2: The last sentence on this page is grammatically incorrect/incomplete.
  • 3, l.91: word order: “96,191 deaths were recorded”
  • 3, l.94: “high variability” – maybe emphasize the three pronounced pandemic waves visible in Figure 1
  • 3, above Fig. 1: “shows the values from Figure 2(a)”
  • 4, unnumbered line below l.105: remove “please”
  • 4, third unnumbered line below l.110: “adequate”
  • 4, l.111: “that the entire period…”
  • 5, ll.113-114: “Other than for the log-exponential model… log-Gompertz models…”
  • 5, l.114: replace “indefinite” by “infinite”
  • General remark: do not capitalize the terms “log”, “exponential” and “logistic”
  • 5, l.118: please clarify that you mean by “peak of the pandemic” the day with the highest number of deaths
  • 5, unnumbered line below l.123: “each period of the pandemic”
  • 5, l.136: “recording”
  • 6, l.147: “registered”
  • 7, l.159: “the models fitted on the 30th and 60th day, respectively, in the log-scale…”
  • 8, l.163: “growth rate” – and also growth type?!
  • 8., ll.166-167: “in the same Cartesian plan[e] of the updated model” – please rephrase
  • 9, l.172: “change point”
  • 9, l.175: “sceneries” => “scenarios”? (also later in the manuscript)
  • 9, l.177: “each of the”
  • 9: Why is equation (3) numbered, whereas other equations do not have numbers? Why does Y_i not have a hat (symbolizing the estimator) unlike in other equations? Similar for equations (4) to (6)
  • 10, l.191: “the parameters”
  • 11, l.197: “the previous section” (better write: “section xx”)
  • 11, ll.199-200: word order – “periods at which the change in the… behavior happened”
  • 11, l.201: word order – “…day the log-Gompertz model for D_2 was maintained”
  • 11, l.205: “representing possible change points”
  • 11, second unnumbered line below l.213: “has the following”
  • 11, l.216: “projection”
  • 11, ll.216-217: “the point estimated for the parameters”
  • 11, l.221: “recorded”
  • 13, l.224: “As another illustration…”
  • 13, l.226: word order – “In this period, 2,903,709 cases of COVID-19 were recorded.”
  • Equation (6): It might be more elegant to use different symbols for (logarithmic) deaths and (logarithmic) cases.
  • Conclusions: I recommend using present perfect instead of simple present when referring to what has been achieved in the manuscript.
  • 14, l.244: “for each updating”?
  • 14, l.246: “each of the sub-datasets”
  • 14, l.247: “seven periods of the pandemic”
  • 14, l.248: “each period… growth models”
  • 15, l.261: word order - “However, if a change point is identified”
  • 15, l.262: “a peak”
  • 15, l.264: “that more restrictive containment strategies need…”
  • 15, l.266: word order – “that a change happened towards a situation”
  • 15, l.267: “it is possible”
  • 15, l.268: “get a flattening”
  • 15, ll.268-270: incomplete sentence
  • 15, l.271: “show that it is”
  • 15, l.273: “growth”
  • 15, l.281: “Y_t” instead of “Y_T”?
  • 15, unnumbered line below l.283: “known… epidemic disease”
  • 15, l.286: “Figure A1 shows the exponential…”
  • 15, l.288: “increases”
  • 15, l.289: replace “indefinitely” by “infinitely”
  • 15, l.291: “is bound, for instance, by the population size”
  • 16, l.298: “Figure A2 shows the logistic…”
  • 16, l.300 and l.301: “symbol”
  • 16, l.301: “needed”
  • 16, l.303: “similar to that”
  • 17, l.308: “ordinate value”
  • 17, l.309: “this value”
  • 17, l.311: “Figure A3 shows the Gompertz…”
  • 17, l.313: word order – “the curve is more inclined”
  • 17, l.313: “symbol”
  • 17, l.314: “needed”
  • References: Please check that (1) either no or all journal names are abbreviated (I suggest none), (2) references [7] and [8] are complete and correctly formatted, (3) the authorship of [15] is corrected, (4) nouns are properly capitalized in the titles of [8] and {14] and the journal name of [20]

Author Response

Please, find attached the point-to-point answers to the raised questions.

Reviewer 3 Report

Unfortunately, the Authors did not convince me with their replies and with the current version of the paper. I still believe that a retrospective fitting of some functions to data does not provide much information and does not help decision makers to fight against the epidemic. The approach completely ignores the dynamics of the epidemic, hence, results obtained from an eariler period of the epidemic do not say much about the situation under changed circumstances. 

Author Response

(The authors gave the same response as above.)

Round 3

Reviewer 1 Report

The authors have properly addressed most of my previous comments in their second revision. . Given this, I feel now that only a few final amendments are necessary before this work can be finally published.

  1. In my previous round of comments, I have raised the issue of the (missing) continuity of the fitted model. The authors answered to this comment in a satisfactory manner in their response letter; however, I would strongly recommend to include this discussion as a short paragraph (e.g., at the end of Section 2) in their manuscript for full transparency of this point.
  2. A remaining (yet minor) statistical problem with the initially fitted “global” models especially in Figure 3 to 6 is that both Y^{(1)} and Y^{(2)} appear to systematically underestimate the initial number of deaths, i.e. they provide a biased (too optimistic) model of the real evolution of deaths. It is still not clear to me where this bias originates from, and this should to be briefly discussed in my opinion. There are a few related statistical points, especially the fact that the residuals are all positive making them serially correlated and, hence, violating the initial assumption of uncorrelated residuals as a key prerequisite for a proper NLS regression model. In their response to my previous comment on this issue, the authors indicate that this problem does not occur anymore in Figures 8, 10 and 11; hence, using piecewise models can be partially motivated by relieving the previous bias. This could (should?) be stated more explicitly in the final manuscript.

Minor comments:

  1. Tables 2 to 7 still need expanded captions referring to the specific model/dataset used. For example, Table 2: “MSE, AIC and BIC values for the candidate models fitted to dataset D_1” or Table 3: “Parameter values for the model \hat{Y}_t^{(1)}”
  2. Page 4, l.100: I suggest to better write here “log-normal distribution with parameters \mu=0 and \sigma^2” and not emphasize the link between \mu and \sigma^2 on one hand, and the mean and variance of the log-normal distribution on the other hand. In fact, both mean and variance of a log-normal distribution are determined by both \mu and \sigma.

Technical suggestions:

  • Page 1, l.5: “of COVID-19”
  • Page 1, l.7: “procedure is capable”
  • Page 1, l.25: “4 million deaths”
  • Page 2, l.48: “this article introduces a piecewise model”
  • Page 2, l.74: “argue that” (optional)
  • Page 1, l.75: “health”
  • Page 2, l.77: “adequate”
  • Page 3, l.89: “to be the number”
  • Page 4, first unnumbered line below l.105: “scale of thousands”
  • Page 4, l.111: “divided into”
  • Page 5, l.115: “log-models”
  • Page 5, ll.116-117: “In the context of COVID-19…”
  • Page 5, l.121: “number of deaths”
  • Page 5, ll.121-122: “growth is characterized by a decreasing rate”
  • Page 5, l.123: “deaths”
  • Page 5, l.136: “assumes”
  • Page 5, l.138: “these two issues”
  • Page 5, l.140: “as model selection criteria instead of the LRT”
  • Page 6, l.167: “Until the 23th day, 496 deaths were registered.”
  • Page 6, l.168: Do you mean “percentage error”?
  • Page 7, l.175: “that we refer to as updated D_1”
  • Page 10, l.213: “Section”
  • Page 10, l.219: Do you mean “percentage error”?
  • Page 11, l.221: “Section”
  • Page 11, l.226: “it was possible”
  • Page 11, l.233: remove “up”
  • Page 11, first unnumbered line after l.238: “14 analysis steps”
  • Page 12, l.239: “analysis step” ´
  • Page 13, unnumbered line after l.244: “there will be additional”
  • Page 13, l.246: You probably mean Table 8 instead of Table 7?
  • Page 13, l.247: “absolute”
  • Page 13, unnumbered line after l.247: Do you mean “percentage error”?
  • Page 13, l.251: “on the 334th day”
  • Page 13, l.252: “on the 364th day”
  • Page 13, l.254: “That is, it happened…”
  • Page 13, l.258: “excessive increase”?
  • Page 13, l.261: “the closing of commerce”
  • Page 14, l.263: “it happened”
  • Page 14, l.264: remove duplicate dot at the end of the sentence
  • Page 15, l.312: “this indicates that a change happened”
  • Page 16, l.335: “increasing the value of \alpha_2 increases the rate…”

Author Response

Dear reviewer ... 

Please find attached our point-by-point answers to the raised comments.

Best regards.

Reviewer 3 Report

Thank you for the updated version and the explanations. Now I think the paper can be accepted for publication. A couple of minor corrections to be performed: "number of death" should be "number of deaths"; line 251: "on the 334-th day"; line 263: " That is, a change to a phase “little better” happened..." 

Author Response

Dear reviewer ... 

Please find attached our point-by-point answers to the raised comments.
